# A large-scale data set of aircraft interaction networks

**Raúl López-Martín[1]\*, Massimiliano Zanin[2]**

**1** Instituto de Física Interdisciplinar y Sistemas Complejos CSIC-UIB, Campus Universitat de les Illes Balears, Palma de Mallorca, Spain, **2** Instituto de Física Interdisciplinar y Sistemas Complejos CSIC-UIB, Edifici Complex de Recerca de les Illes Balears, Parc Bit, Palma de Mallorca, Spain

\* raullopez@ifisc.uib-csic.es

## Abstract

Complex network concepts have become the foundation of many real-world studies, encompassing topics like the dynamics of spreading processes or the evaluation of the resilience of complex systems. One of their major enablers is the availability of real data sets, on which to test and validate models and algorithms. We here present a data set containing 1,708 network representations of daily interactions between aircraft over a vast area of the European airspace, for 854 days spanning between 2015 and 2021. It has been obtained by processing trajectories, both planned and executed, and comparing aircraft positions in a pairwise fashion. This is further complemented by metadata about the status of the airspace, in a multi-scale structure. This database may act as the data source of any study willing to use these interactions to develop new tools for understanding air traffic dynamics; and more generally, to test complex networks algorithms and models on large-scale real graphs.

## Introduction

Spreading phenomena in networks are a pervasive topic, with applications in many natural, social and technological contexts [1–5]. Sometimes the aim is to disrupt the propagation, for instance of contagious diseases in a social network; sometimes we want to enhance it, e.g. to facilitate the distribution of information or resources among the participants in a market; or we may just intend to understand the propagation process and the mechanisms supporting it, as is the case of understanding gene-cellular networks. In all cases, two elements are essential: algorithms and methods for modelling the dynamics and designing interventions; and real data sets on which to test and validate these methods. Since the inception of complex networks theory [6,7], it has been realised that the topology of the network has a major impact on the spreading dynamics [8–12]; hence, while synthetic models are still useful to test specific hypotheses, reliable analyses can only be supported by real data sets.

We here contribute a curated data set representing interactions between aircraft through complex network structures. Starting from a large collection of planned and

**Data availability statement:** The data files underlying the results presented in the study are available from the Zenodo database: https://zenodo.org/records/15017762.

**Funding:** This project has received funding from the European Research Council (ERC) under the European Union's Horizon 2020 research and innovation programme (grant agreement No. 851255). This work was partially supported by the María de Maeztu project CEX2021-001164-M funded by the MICIU/AEI/10.13039/501100011033. R. L.-M. acknowledges support from the Spanish Ministry of Science, Innovation and Universities through the grant FPU22/03765. There was no additional external funding received for this study. The funders had not role in the design, data collection and analysis, decision to publish, or preparation of the manuscript.

**Competing interests:** The authors have declared that no competing interests exist.

executed (radar) trajectories, we here model these as networks, in which nodes represent individual aircraft, pairwise connected when their distance fell below a threshold. Links therefore represent instances in which aircraft interacted, as e.g. their trajectories had to be changed to avoid possible safety issues, or more in general, when they attracted the attention of air traffic controllers. Two networks are provided per day, i.e. one for planned and one for executed flights; including four months per year, from 2015 to 2021. This data set has previously been used to understand the structure created by interactions throughout the whole European airspace through different complex network metrics [13–15], with the aim of unveiling the factors affecting the complexity of air traffic. The daily evolution of the structure was found to be dependent on the traffic volume, especially under strong perturbations - e.g. during the COVID-19 pandemic [14]. The topology was also found to have a multi-scale structure, reflecting the internal organisation of the airspace [15].

Compared to other real-world networks available for research purposes [16–18], the ones here shared present several characteristics that are both opportunities and challenges. Their topology is highly non-trivial, evolving according to weekly and yearly seasonalities; is highly modular; and with the node degrees partially following a power-law (see also results below). Networks are embedded in a three-dimensional space: links (i.e. interactions) take place at specific locations, and the corresponding nodes (i.e. flights) move across the airspace. They also have a multiscale temporal nature: while individual interactions take place in a scale of seconds, chains of interactions (i.e. paths in the network) can span hours; consequently, they can naturally be interpreted as time-evolving networks. Interactions are associated to different intensities, representing the minimum horizontal separation recorded between two aircraft. Finally, as separated networks are provided for each day, the full data set comprises 1,708 instances with similar, albeit not equal, structure, thus providing a natural source of variability. Note that this represents a size comparable to some of the largest available network repositories [16–18].

The contributed data set also represents a major shift in the context of air traffic management. To the best of our knowledge, the only other publicly-accessible data set providing similar information was presented in Ref [19], and covers approximately 11 months of 2022 and flights crossing the Air Control Center of Bordeaux (LFBBDX), France. In contrast, the data set here presented is the first instance spanning a full continent and multiple years; this supports an analysis of the system from a macro-scale level and capturing its wide variance, including the yearly and seasonal variations, and the differences between control regions.

The contributed networks find a natural application in many problems within the context of air traffic management and control. Firstly, this data set can be used to validate existing methodologies, as e.g. the one presented in Ref [20], whose core idea is to reconstruct temporal interaction networks and study the propagation process between flights using a SIS epidemiological model [21]. Secondly, our networks include a wide range of topologies, i.e. interaction scenarios, which can be used to understand factors contributing to Air Traffic Control Officers (ATCOs) workload [22, 23]; and to assess Conflict Detection and Resolution (CD&R) algorithms [24]. Finally, these networks can be used to describe the air traffic dynamics through time

and space, and at different levels of granularity [13–15]. At the same time, this data set can be used in many additional network-related topics. Networks can be seen as the structure supporting a real propagation process, and as such can be used to test algorithms to disrupt (or enhance) such dynamics; for instance, researchers can use them to test the effectiveness of network dismantling algorithms [25,26], with the advantage of providing a large number (1,708 networks) of individually large (an average of 22 thousand nodes and 49 thousand links) instances. These networks can also be used as test-bed for other related problems, as for instance link prediction [27–29], community identification [30,31], representation of higher-order interactions [32,33], or testing Graph Neural Network models [34,35].

## Materials and methods

### Raw trajectory data pre-processing

Original data of aircraft operations were obtained from the EUROCONTROL's R&D Data Archive, a public repository of historical flights made available for research purposes and freely accessible for the academic community, subject to users agreeing the terms and conditions [36]. The data set includes information on all commercial and general aviation flights (i.e. excluding sensitive, state, and military flights) operating within and over Europe, incorporating flight plans, radar data, and the associated airspace structure. Data availability is constrained at the source to four months - March, June, September, and December. We further consider seven years available at the time of accessing the data set, from 2015 to 2021 (both included). For each day, the executed and planned trajectories were extracted for each flight landing in that specific day. The planned trajectories are reconstructed using the flight plans submitted by airlines and other aircraft operators to EUROCONTROL's Network Manager (NM), and further updated with data from EUROCONTROL's Central Route Charges Office (CRCO). As such, planned trajectories are not necessarily reflecting the initial intentions of airlines, which may have been modified according to capacity restrictions and other operational limitations. On the other hand, executed trajectories are reconstructed according to radar observations of the flight's path. The average temporal resolution, measured for June $1^{st}$, 2019 as the time between consecutive position reports, is of 278.4 s for planned trajectories (standard deviation of 236.5 s), and of 282.6 s for executed ones (standard deviation of 230.9 s).

Several pre-processing steps are performed on these trajectories - see the top part of Fig 1 for a graphical representation. First, trajectories described by four or less points are discarded. A manual inspection revealed that these mainly corresponded to helicopter movements - note that, while helicopters are not the main scope of this data set, their trajectories are included when they involve the use of controlled airspace, and may therefore interact with other flights.

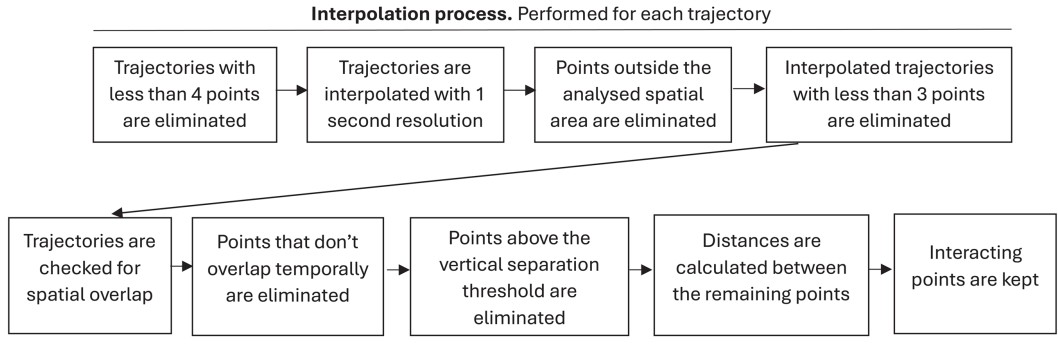

**Interpolation process.** Performed for each trajectory

Trajectories with less than 4 points are eliminated → Trajectories are interpolated with 1 second resolution → Points outside the analysed spatial area are eliminated → Interpolated trajectories with less than 3 points are eliminated

Trajectories are checked for spatial overlap → Points that don't overlap temporally are eliminated → Points above the vertical separation threshold are eliminated → Distances are calculated between the remaining points → Interacting points are kept

**Interaction tracking process.** Performed for each pair of interpolated trajectories

**Fig 1. Flowchart of the trajectory pre-processing and interaction extraction.** The main steps are summarised, with the chronological sequencing represented by arrows. The top part includes the processes associated to the interpolation, while the bottom one those associated to the extraction of interactions. Each set of processes has been performed for each day and each type of trajectory presented in the dataset.

We additionally identified a small set of very short flights, e.g. functional check flights, performed following maintenance actions on the aircraft to confirm its airworthiness; short flights between secondary airports; and flights for which no trajectory was reported. In most of these cases, the information available in the planned and executed trajectories was similar, confirming that these are genuine examples of short or technical flights, and not the result of data processing errors. In total, these instances represented a minimum share of the whole data set - e.g. approximately 0.28% of all flights for June 28$^{th}$, 2019.

Next, in order to homogenise the temporal resolution of trajectories, a linear interpolation is performed on the three spatial dimensions between each available position report, to reach a resolution of one point per second. No additional smoothing or noise reduction technique has been applied. After this interpolation, only those points falling within a simplified European airspace are retained, defined by the geographical rectangle included between $-15°$ and $30°$ in longitude, and between $35°$ and $70°$ in latitude. Any trajectory left with less than three points within this region is discarded. These corresponded to a 4.7% of the total flights for June 28$^{th}$, 2019; and in all cases, corresponded to flights crossing the European airspace outside the rectangular boundaries previously defined.

Finally, all trajectory points whose altitude was below 100FL (Flight Levels, or 10,000 feet) were removed; as have been all flights not reaching such altitude. This latter filter only affects general aviation flights and very short trajectories. Note that the use of this altitude threshold is motivated by two operational considerations. Firstly, the quality of the trajectory data is higher while en route, as aircraft have a more constant and predictable dynamics. Secondly, operations near airports may comprise heterogeneous complex route structures, in which a reduction of the distance between two aircraft is part of the intended departure or arrival procedure - one may think, for instance, of the simultaneous landing of two aircraft in parallel runways.

For the sake of completeness, Table 1 reports an overview of the number of flights removed in each step for June 28$^{th}$, 2019, i.e. the day with most flights in the data set, for both executed and planned operations. Note that these trajectories, due to their short length, did not participate in a substantial number of interactions; to illustrate, when they are not deleted, the number of detected daily interactions only increases, on average, by 0.871 for planned trajectories (standard deviation of 1.49, maximum of 14), and by 0.166 for executed ones (standard deviation of 0.550, maximum of 6).

### Extraction of interactions

In the context of this work, an interaction between two flights represents an instance in which their reciprocal horizontal distance falls below a threshold of 10NM, while the vertical one simultaneously falls below 2,000ft. Note that such situations do not necessarily imply a safety-critical condition. On the one hand, intersections of planned trajectories are accepted, as those trajectories may only be modified to comply with capacity limitations of airspaces and airports. On the other hand, 10NM is well above the minimum distance for maintaining a safe separation. On the contrary, these interactions can be understood as situations in which air traffic controllers have to start paying attention to the pair, and eventually take resolutory actions. At the same time, these interactions can be seen as a propagation process: solving an interaction by changing the trajectory of one (or both) aircraft can result in the creation of a later interaction with a third flight - something known as a downstream effect [37].

For each pair of aircraft in the same day and in the same data set (i.e. either planned or executed), interactions are calculated by checking the minimum horizontal and vertical distance they achieve, across the full duration of both flights.

**Table 1**. Information about number of flights deleted during the pre-processing of the trajectory data, for June 28$^{th}$, 2019.

| Trajectory type | # flights | # pre-interpolation deleted flights | # post-interpolation deleted flights | # sub FL100 flights |
|---|---|---|---|---|
| Executed | 34421 | 98 | 1618 | 499 |
| Planned | 34421 | 117 | 1610 | 443 |

Whenever both separations are below the corresponding thresholds, an event is recorded. In case of multiple interactions between the same couple of aircraft, only the one corresponding to the minimum horizontal separation is retained. Additional information that is stored for each interaction include its time stamp, the geographical position, and the minimum horizontal distance.

A synthetic example of the reconstruction process is depicted in Fig 2, for four temporal snapshots (from left to right, and top to bottom). For the sake of simplicity, in this example aircraft are assumed to fly at the same altitude; the vertical position of aircraft, and hence the vertical separation, are thus neglected. Two aircraft, 1 and 2, with intersecting trajectories have an initial horizontal separation above the threshold (green line, top left), and enter the interaction range in the second snapshot (orange line). Note that a link is yet not created between the corresponding nodes in the network, as this is added only when the minimum horizontal distance is achieved (red line, bottom left panel). Due to the possibility of an unsafe event, the air traffic controller has to change the original trajectories (see the red dashed lines), leading to a new interaction between aircraft 1 and 3 in the bottom right panel - which would not have happened, had the trajectory of aircraft 1 not been changed. The final result is the network represented below the fourth panel, with three nodes and two links, in addition to the time stamp of the moment in which the minimal separation is achieved.

The interaction extraction process, also depicted in the bottom part of Fig 1, is repeated for all pairs of trajectories, i.e. approximately $8.88 \times 10^{12}$ pairs across all days. A total of $8.37 \times 10^8$ interactions were detected across the 854 analysed days, with an average of $4.9 \times 10^4$ and a standard deviation of $2.8 \times 10^4$ per day. Note that the large standard deviation

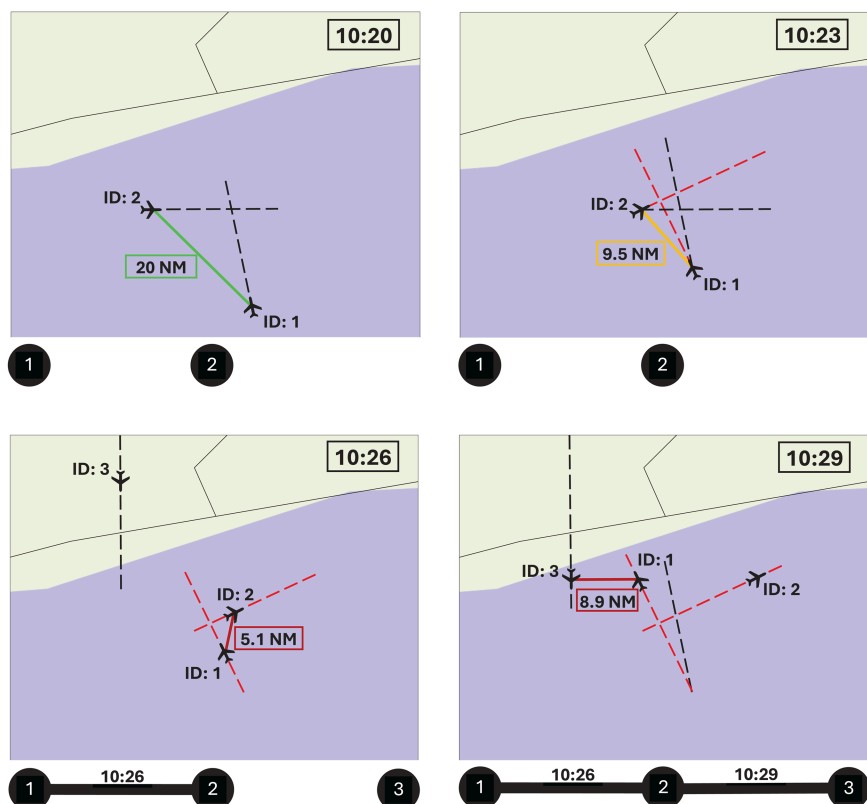

**Fig 2. Temporal snapshots of a simulated interaction network reconstruction.** The four panels, from left to right and top to bottom, represent four different moments in time - see the hour in the top right corner. Black and red dashed lines respectively represent the original and modified trajectories; and solid lines depict the distance between pairs of aircraft. The reconstructed network is reported underneath each panel. See main text for details.

is mainly due to the inclusion of flights in years 2020 and 2021, i.e. when air transport was impacted by the COVID-19 pandemic [38].

## Results

### Data records

All data described below are accessible at https://doi.org/10.5281/zenodo.15017762.

Interaction data are provided in two Comma Separated Values (CSV) files for each day, respectively corresponding to planned and executed trajectories. File names encode the type and date, following the structure *interactions_yyyymmdd_type.csv*, with *type* being either "executed" or "planned". Information in each file is organised in six columns, with each row corresponding to an individual interaction; rows are further sorted in increasing order of the time at which the interaction took place. The first two columns encode an ID of the interacting flights; note that this number is randomised each day (i.e. flight *a* of one day is not the same flight on another day), but the identity of a flight between planned and executed files is maintained. Interactions are encoded per flight, and not per aircraft; in other words, one same aircraft may operate multiple flights throughout one day, but these are considered as separate entities (i.e. separate nodes). Next, the third column encodes the time at which the interaction occurred, in seconds starting from the time of the first interaction of that day. Fourth, we include the minimum horizontal separation achieved by the two interacting aircraft, in nautical miles; this metric contributes to the severity of the event and can thus support the creation of a weighted network. The last two columns report the location where the event took place, including the barometric standard altitude (in tens of Flight Levels, or multiples of thousand feet, fifth column) and the associated FIR (Flight Information Region, i.e. one basic level of division of the airspace, sixth column). The latter is also randomly codified, but the coding is maintained constant across days, to support comparisons through time. For the sake of clarity, Table 2 reports a synthesis of the meaning of columns; Table 3 an example of the data contained in the file 2019-06-28 for executed trajectories; finally, Fig 3 reports a graphical representation of the sub-network corresponding to a FIR for that day.

We complement the above information with statistics about the overall traffic situation for each day. These are provided in two additional CSV files for each of the 854 days, i.e. one for each type of trajectory analysed; and contain a set of macroscale variables describing both the entire European airspace under consideration and each considered FIR. File names follow the previous format, i.e. *metadata_yyyymmdd_type.csv*. Six different metrics are included. The first column defines the area or FIR for which the metadata are given; note that the anonymised ID of the FIR is used, that is, the

**Table 2**. Structure of the CSV files with information about interactions.

| Column # | Column name | Information | Format |
|---|---|---|---|
| 1 | flId1 | Identifier of the first involved flight. | Integer |
| 2 | flId2 | Identifier of the second involved flight. | Integer |
| 3 | time | Time since the first interaction, in seconds. | Integer |
| 4 | distance | Minimum horizontal separation, in Nautical Miles. | Real |
| 5 | altitude | Barometric standard altitude of the event, in tens of Flight Levels. | Integer |
| 6 | firId | Identifier of the FIR in which the interaction took place. | Integer |

**Table 3**. Initial five rows from the interaction CSV file for executed trajectories of 2019-06-28. Distances are rounded to $10^{-3}$.

| flId1 | flId2 | time | distance | altitude | firId |
|---|---|---|---|---|---|
| 13206 | 517 | 0 | 6.025 | 340 | 43 |
| 26020 | 31530 | 1322 | 9.177 | 320 | 33 |
| 28941 | 19011 | 4050 | 4.764 | 330 | 3 |
| 10908 | 7270 | 5288 | 7.379 | 340 | 85 |
| 4255 | 13088 | 7211 | 9.208 | 300 | 85 |

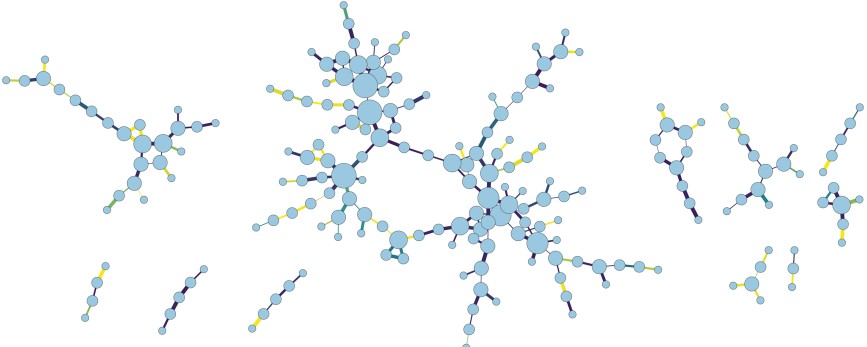

**Fig 3**. **Graphical representation of the main components of the network of interactions in a single FIR for June 28$^{th}$, 2019.** The size of nodes is proportional to their degree; the colour of links to the altitude at which the interaction took place, from yellow (FL100) to dark blue (FL400), and their thickness to the minimum horizontal distance reached.

same as in the interactions' files, with an additional value "-1" representing the total considered European airspace. The second, third and fourth columns respectively report the total number of flights that have flown over the corresponding section of airspace; the total flown distance, in nautical miles; and the total flown time, in seconds. The two last columns contain a measure of the spatial entropy of the trajectories, i.e. of how heterogeneously they are spread in the airspace, thus representing the complexity of the traffic from a control viewpoint [15]. Qualitatively, this entropy is calculated by dividing the sector in a grid of cells, and computing the amount of times throughout the day each cell is crossed by a flight. This result is translated into a probability distribution, and the entropy is obtained by applying Shannon's entropy formula [39] over such distribution. The second entropy value corresponds to a normalisation by area, according to the maximum entropy such airspace can have (details about these two metrics are available in Ref [15]). As for the other set of data files, Table 4 reports a synthesis of the meaning of columns; and Table 5 an example of the data contained in the file 2019-06-28 for executed trajectories.

**Table 4**. **Structure of the metadata CSV files.**

| Column # | Column name | Information | Format |
|---|---|---|---|
| 1 | firId | Identifier of the FIR, i.e. of the section of airspace; "-1" represents the aggregated considered European airspace. | Integer |
| 2 | numFlights | Number of flights over the area. | Integer |
| 3 | distance | Distance flown over the area, in Nautical Miles. | Real |
| 4 | time | Time flown over the area, in seconds. | Integer |
| 5 | entropy | Entropy of the trajectories over the area. | Real |
| 6 | normEntropy | Entropy of the trajectories, normalised by the area of the section. | Real |

**Table 5**. **Initial five rows from the metadata CSV file for executed trajectories of 2019-06-28. Distances and entropies are rounded to $10^{-3}$.**

| firId | numFlights | distance | time | entropy | normEntropy |
|---|---|---|---|---|---|
| -1 | 32206 | 11967204.887 | 155645278 | 14.557 | 0.878 |
| 0 | 2799 | 153258.751 | 2937931 | 10.017 | 0.825 |
| 1 | 1727 | 310195.641 | 4166942 | 11.845 | 0.847 |
| 2 | 711 | 96238.745 | 880762 | 9.754 | 0.835 |
| 3 | 1573 | 241443.334 | 1814991 | 11.216 | 0.829 |

As a final note, for the sake of easy access and download, all files related to a type of trajectory and a given month, alongside the corresponding metadata, are stored in a single ZIP file. Additional information about individual events can be requested to the authors, prior proof of registration to the EUROCONTROL's R&D Data Archive.

## Technical validation

The validation of the interactions reported in this data set is a challenging task, mainly due to the lack of similar data sets that could be used for comparison; even statistics about the appearance of aircraft interactions are scant, due to their sensitive nature. Quality assurance has thus been performed by resorting to a three-fold approach.

Firstly, all steps in the preparation of the data (e.g. trajectory filtering, interpolation, and interaction detection) has been independently developed by one of the authors (R.L.-M.) and tested by the other (M.Z.). Such tests have been conducted using a Unit Testing approach. The most salient outputs have manually been inspected, including the flights that have been deleted in the pre-processing phase due to lack of data (see Table 1).

Secondly, we have compared the number of detected interactions in executed trajectories, with the number of separation losses (or Separation Minima Infringements, SMI) officially reported each year. This latter number includes all events with a safety implication in which two aircraft came too close. Note that the concept of interaction here considered is wider in scope, due to the larger distance thresholds that have been used, and therefore include both safety critical and non-safety critical events. The number of SMIs has been extracted for each Functional Air Block (FAB) in the European airspace between years 2015 and 2019, from the corresponding Annual Monitoring Report prepared by the European Aviation Safety Agency (EASA) in support to the Performance Review Body (PRB) of the Single European Sky (SES) - available at https://eu-single-sky.transport.ec.europa.eu. We finally calculated a linear correlation between the evolution of both sets of values, obtaining $R^2$ of 0.959 ($p$-value of $4 \cdot 10^{-25}$). It can then be concluded that, while our definition of interaction is wider by design, and includes on average 2,400 events for each SMI, both concepts are correlated and no anomalous trend is present.

Thirdly, the output of the analysis, including the CSV files here provided, has been tested for coherence by calculating some basic statistics. Specifically, Fig 4 depicts the spatial and temporal distributions of interactions for June $1^{st}$, 2019. It can be appreciated that interactions most frequently appear at the intersection of airways and in busy airspaces, and at hours of the day with most traffic, as is to be expected. Additionally, most interactions appear in the enroute phase (with altitudes above FL350, i.e. where aircraft spend most of the time).

Moving to the network structure, Fig 5 depicts the evolution through time of six classical topological metrics [40], calculated for all days in March and June 2019, and including: the average degree, number of links and nodes, fraction of isolated nodes, modularity (estimated using the Louvain algorithm [41]), and weak giant cluster size. On the other hand, Fig 6 reports the complementary cumulative distribution functions (CCDF) of the number of links, horizontal separation, and harmonic centrality [42]. The final panel of the same figure finally reports the evolution of the network's efficiency under a pruning process, in which a random link connected to the node with the highest degree is iteratively removed. In the two latter figures, the evolution of the network structure reflects known periodicity and trends in traffic volumes.

## Analysis of daily interaction networks

To show an example of a potential application of this data set in an air traffic management context, we here present an analysis of the structure of interaction networks by day. This extends what initially presented in Ref [14], by focusing on the dependence of the structure on the total distance flown. Fig 7 presents the values of four different topological metrics for each available day between years 2015 and 2019, i.e. before the COVID-19 pandemic, as a function of the total distance flown over Europe by all aircraft in the corresponding day. On the one hand, the first thing that can be seen are clear and different tendencies for each metric. The degree entropy, weak giant cluster size and efficiency increase with the total distance flown; meaning that the network becomes both more heterogeneous and better connected. Note that

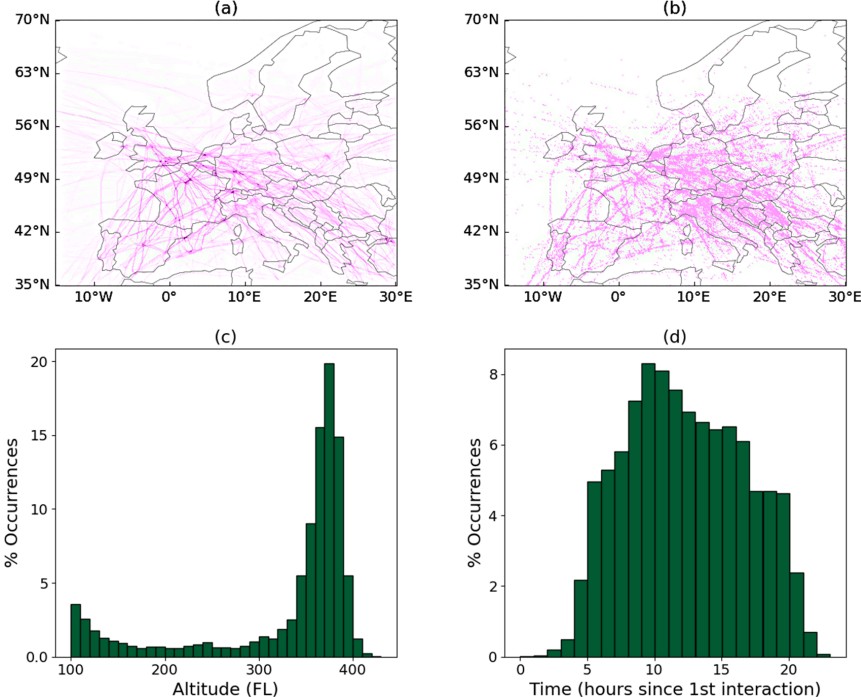

**Fig 4**. **Spatial and temporal distribution of interactions.** Panels (a) and (b) respectively show the spatial locations of planned trajectories, and (b) of the corresponding interactions. Darker shades indicate higher densities of trajectories and interactions. Panels (c) and (d) depict histograms of the altitude and time of the day at which interactions took place. In all cases, data correspond to June $1^{st}$, 2019.

this is a negative feature from the controllers' perspective, as it implies both that more aircraft interact, and that some of them are involved in potentially many conflicts. This tendency is further confirmed by a decrease on the ratio of isolated nodes as the distance increases, implying that a larger percentage of aircraft takes part in interaction events. On the other hand, a more surprising result is obtained when considering differences across days of the week - see the colour of points, and the figure's legend. While most of the weekdays strongly overlap, Saturdays and (to a less degree) Sundays suffer a change in the offset. We hypothesise that this may be the consequence of a shift in the main source-destination pairs during weekends, during which traffic is changed by the higher demand for touristic destinations. In short, this analysis shows how the influence of the day of the week on air traffic dynamics, and hence on the interactions appearing between aircraft, can be illustrated and quantified through the structure of the corresponding network representation. The interested reader can find extended discussions in Refs. [14,15].

## Computational cost

One important aspect of the analysis of these networks is the corresponding computational cost, as, due both to their size and their temporal nature, such cost can become significant.

The left panel of Fig 8 firstly report the time required to create the described networks, as a function of the number of aircraft to be tracked. These values have been obtained by starting with the planned trajectories for June $28^{th}$, 2019, i.e. the day with the highest number of inflights; and artificially deleting aircraft at random, to simulate smaller data sets. It can be appreciated that the cost scales almost linearly with the number of aircraft; this is due to the fact that, while the number of interactions scales quadratically, most pairs of aircraft cannot physically interact (e.g. they operate at different hours), and are thus not checked.

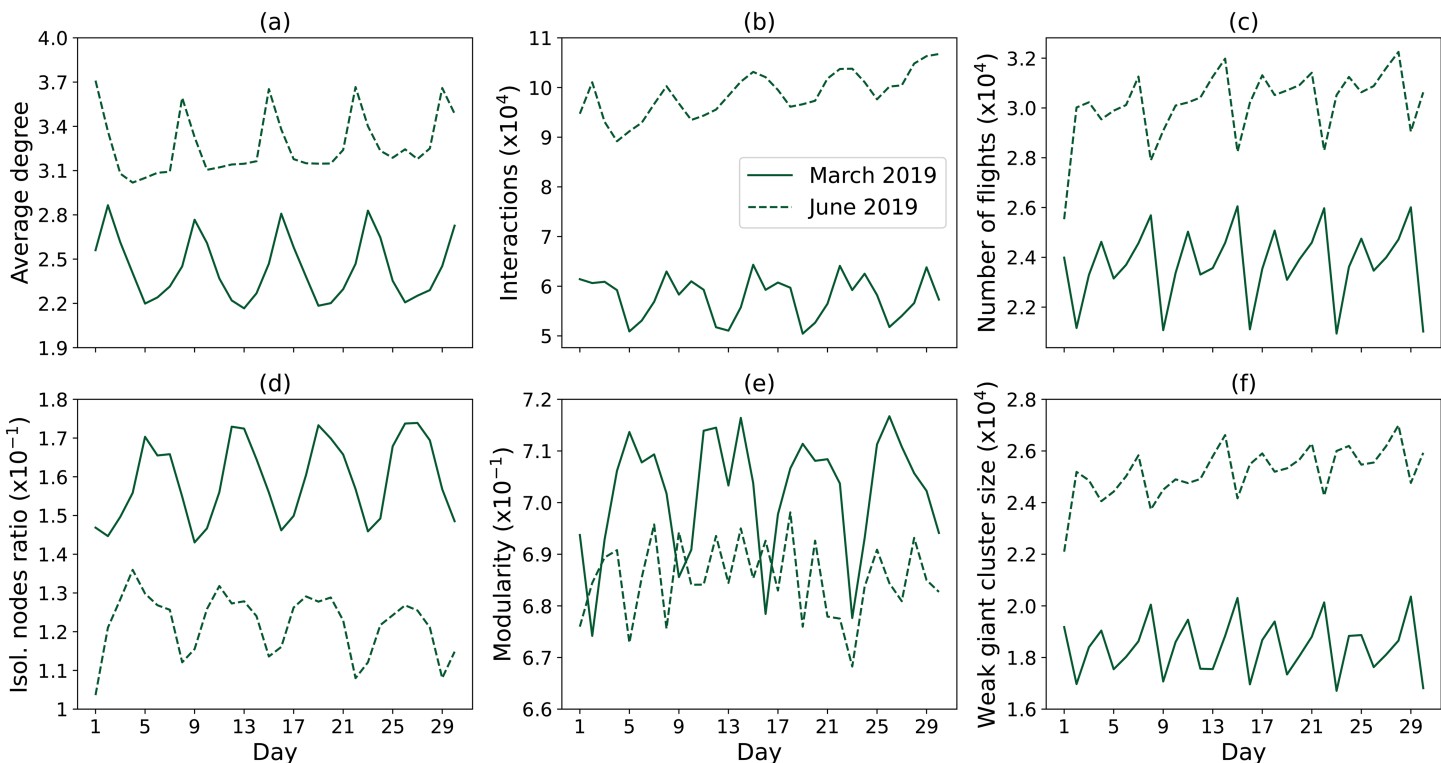

**Fig 5**. **Time series of complex networks measurements extracted from interactions for planned trajectories over the whole European airspace.** These include: (a) the daily average degree; (b) the number of interactions, i.e. of links; (c) the number of flights, i.e. of nodes; (d) the ratio of isolated nodes; (e) the modularity; and (f) the weak giant cluster size. Data correspond to March 2019 (solid lines) and June 2019 (dashed lines).

Next, the central and right panels of Fig 8 report the time required to calculate two classical topological metrics on the resulting networks, namely the modularity (estimated using the Louvain algorithm [41]) and the global efficiency, as a function of the number of interactions in the network.

## Usage notes

While the use of this data set will strongly depend on the specific application being tackled, we include a set of files designed to illustrate how to load and perform some basic operations on the data. Each program is coded in Python, using only standard libraries, and includes a basic set of comments to explain its behaviour. The six provided examples include: plotting the daily average degree of nodes, number of interactions, number of flights, and size of the weak giant cluster; the full degree distribution of nodes; and the histogram of the minimum horizontal separation between interacting aircraft. These programs thus allow to reproduce the main topological analyses presented in this contribution, and can be downloaded alongside the data at https://doi.org/10.5281/zenodo.15017762.

## Update

Data for additional years will be included whenever the corresponding trajectory data are made available in the EUROCONTROL's R&D Data Archive, and will appear as linked data sets.

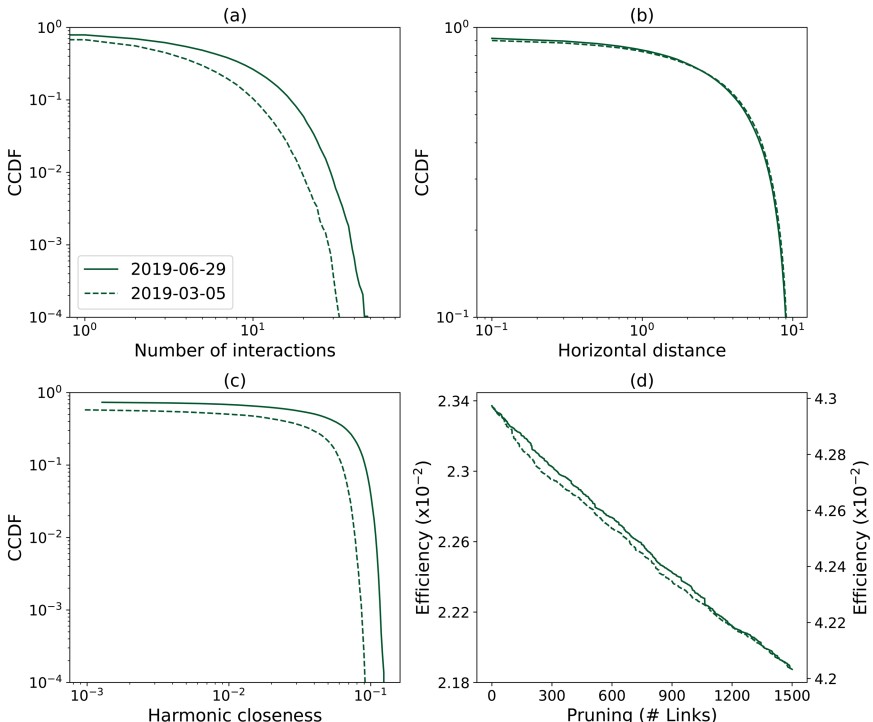

**Fig 6**. **Topological properties of individual networks.** These include: (a) the complementary cumulative distribution function (CCDF) of the degree distribution; (b) the CCDF of the minimum horizontal distance between interacting aircraft; (c) The CCDF of the harmonic centrality; and (d) the evolution of the network's efficiency under a pruning process. Dashed and solid lines respectively correspond to planned trajectories of March 5th, 2019, and June 29th 2019.

## Discussion and conclusions

In this contribution we presented a data set comprising 1,708 temporal, spatially-embedded, and multi-level networks representing aircraft interactions over Europe for 854 days between 2015 and 2021. This collection is freely accessible at https://doi.org/10.5281/zenodo.15017762; and includes additional information about individual events and the global status of each FIR. While similar data sets already exist (see for instance Ref [19]), to the best of our knowledge this is the first instance covering extensive spatial and temporal scales. This can represent a paradigm shift in air traffic research, allowing to capturing the nuances of air traffic dynamics throughout broader time and space scales. Complex networks practitioners at large can also benefit from it, as these networks display a non-trivial, multi-scale and temporally evolving topology. They can thus support the validation of new algorithms and methods, especially in the context of spreading processes.

In spite of many advantages, the interested user should also be aware of the limitations that this data set entails. The reliability of the detected interactions is necessarily a direct function of the quality of the original data, i.e. of the raw trajectories, as provided by EUROCONTROL. While steps have been taken to enhance the quality of the results, as e.g. the trajectory interpolation procedure and the deletion of flights with an unreliably low amount of points, the presence of false positives and false negatives cannot be excluded.

Future improvements of this data set will be directed in two main directions. Firstly, new networks will be added, conditioned to the publication of new raw data in the EUROCONTROL's R&D Data Archive. Secondly, complementary data sources will be used to extract additional factors associated to the interactions, especially related to the workload these cause for air traffic controllers.

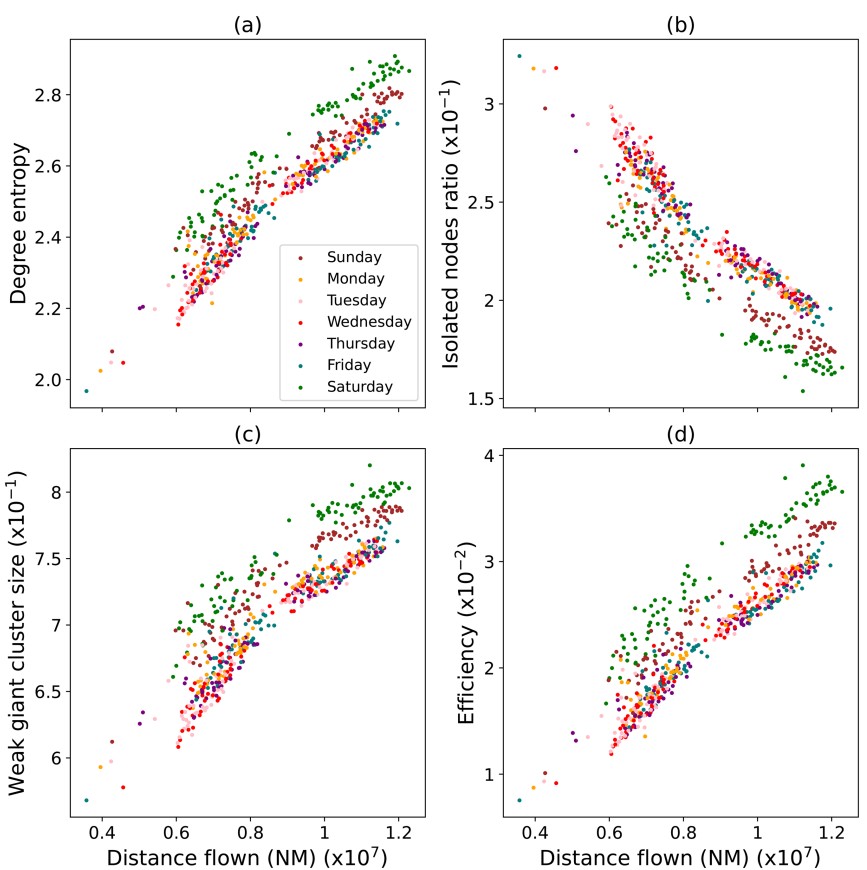

**Fig 7**. **Evolution of daily network metrics as a function of the total distance flown.** From left to right, top to bottom, these include: (a) the degree entropy; (b) the ratio of isolated nodes; (c) the weak giant cluster size and (d) the efficiency. Networks obtained from executed trajectories for all available days ranging from March 1$^{st}$ 2015 to December 31$^{st}$ 2019. The colour of each marker represents the day of the week, see legend in the first panel.

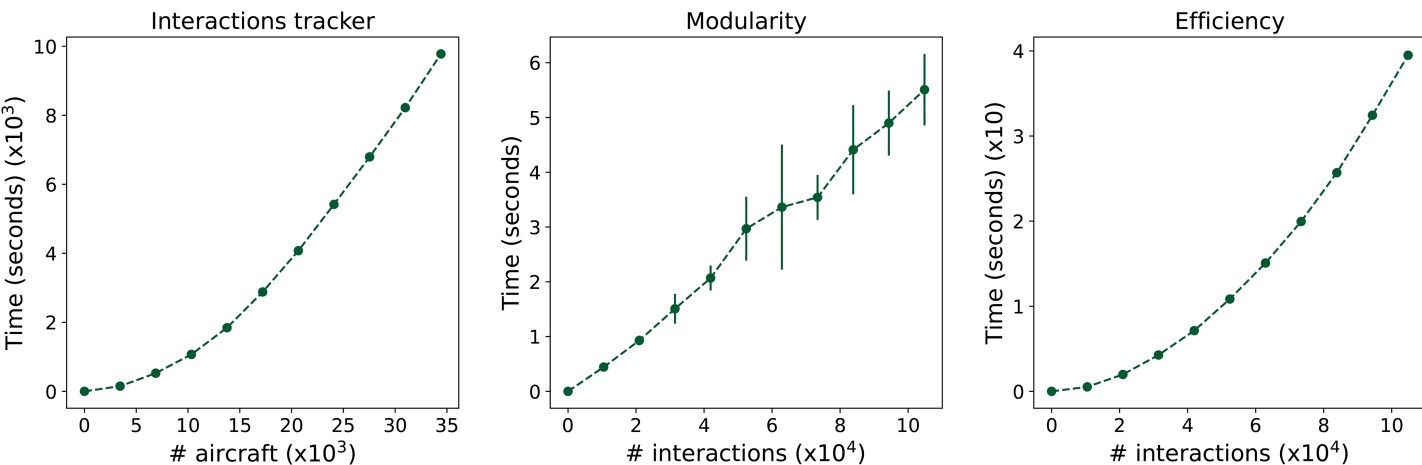

**Fig 8**. **Computational cost of the analyses.** (Left) Time required to reconstruct the interaction network, as a function of the number of aircraft in the data set. (Centre and right) Time required to calculate the modularity and efficiency of the resulting networks, as a function of the number of interactions. Points represent the median, and whiskers the standard deviation over ten independent realisations. In all cases, calculations have been performed using AMD Epyc 7402 processors (limited to a single core); 80 GB of memory was allocated for the interaction tracking algorithm, 4 GB was allocated for the complex network metrics calculation.

## Code availability

Data processing and visualisation were performed using Python 3.9 and standard libraries. Additional Python scripts are further provided alongside the data set to recover the main results of Figs 5 and 6.

## Acknowledgments

This document has been created with or contains elements of ATM Datasets made available by EUROCONTROL ( 2020, EUROCONTROL). EUROCONTROL does not necessarily support and/or endorse the conclusion of this document. EUROCONTROL shall not be liable for any direct, indirect, incidental or consequential damages arising out of or in connection with this document and/or underlying the ATM Datasets.

## Author contributions

**Conceptualization:** Massimiliano Zanin.

**Data curation:** Raúl López-Martín.

**Formal analysis:** Raúl López-Martín.

**Methodology:** Raúl López-Martín, Massimiliano Zanin.

**Software:** Raúl López-Martín.

**Supervision:** Massimiliano Zanin.

**Validation:** Massimiliano Zanin.

**Writing – original draft:** Raúl López-Martín, Massimiliano Zanin.

**Writing – review & editing:** Raúl López-Martín, Massimiliano Zanin.

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
