## [Decision Letter · Decision Letter 0]

11 Sep 2025

PONE-D-25-41897A large-scale data set of aircraft interaction networks.PLOS ONE

Dear Dr. López-Martín,

Thank you for submitting your manuscript to PLOS ONE. After careful consideration, we feel that it has merit but does not fully meet PLOS ONE’s publication criteria as it currently stands. Therefore, we invite you to submit a revised version of the manuscript that addresses the points raised during the review process.

**ACADEMIC EDITOR:**

In light of the reviewers’ evaluations, I am requesting a **major revision**. The reviewers’ suggestions are constructive and intended to strengthen methodological transparency, data validation, and demonstration of the dataset’s utility. Addressing these points will greatly improve the manuscript’s clarity, reproducibility, and impact.

I invite you to revise the manuscript accordingly and provide a detailed response letter outlining how each comment has been addressed. 

We look forward to receiving your revised manuscript.

Kind regards,

Absalom El-Shamir Ezugwu, Ph.D

Academic Editor

PLOS ONE

Journal Requirements:

4. Thank you for stating in your Funding Statement: [This project has received funding from the European Research Council (ERC) under the European Union's Horizon 2020 research and innovation programme (grant agreement No 851255). This work was partially supported by the María de Maeztu project CEX2021-001164-M funded by the MICIU/AEI/10.13039/501100011033. R. L.-M. acknowledges support from the Spanish Ministry of Science, Innovation and Universities through the grant FPU22/03765.].

5.Thank you for stating the following financial disclosure: [This project has received funding from the European Research Council (ERC) under the European Union's Horizon 2020 research and innovation programme (grant agreement No 851255). This work was partially supported by the María de Maeztu project CEX2021-001164-M funded by the MICIU/AEI/10.13039/501100011033. R. L.-M. acknowledges support from the Spanish Ministry of Science, Innovation and Universities through the grant FPU22/03765.]. 

Reviewers' comments:

Reviewer's Responses to Questions

**Comments to the Author**

1. Is the manuscript technically sound, and do the data support the conclusions?

Reviewer #1: Yes

Reviewer #2: Partly

Reviewer #3: Yes

2. Has the statistical analysis been performed appropriately and rigorously?

Reviewer #1: No

Reviewer #2: N/A

Reviewer #3: Yes

3. Have the authors made all data underlying the findings in their manuscript fully available?

Reviewer #1: Yes

Reviewer #2: No

Reviewer #3: No

4. Is the manuscript presented in an intelligible fashion and written in standard English?

Reviewer #1: Yes

Reviewer #2: Yes

Reviewer #3: Yes

5. Review Comments to the Author

Reviewer #1: The paper introduces a unique, large-scale dataset of aircraft interaction networks spanning multiple years (2015–2021) and across the entire European airspace, which is unprecedented in scope. Though there are few points authors should address:

1) While the dataset’s utility is clear, the introduction could better highlight specific gaps in existing resources that this dataset uniquely fills.

2) Why the data is restricted to only four months?

3) Whether the restriction to only four months per year won’t bias results. The implications of this sampling choice could be discussed more explicitly.

4) Proper qualitative, quantitative validation of the data should be provided including some visual presentation.

5) The paper could include a brief demonstration of a real-world analysis workflow such as building a temporal network analysis from raw CSVs.

Reviewer #2: General Summary of the Paper

This manuscript presents an extensive dataset comprising 1,708 animated networks representing daily interactions between aircraft over European airspace spanning from 2015 to 2021. The dataset is generated from processing trajectory data to capture pairwise aircraft interactions, thereby enabling large-scale analysis of air traffic dynamics. The authors aim to provide a publicly accessible resource to facilitate research in complex networks, air traffic management, and related fields. They also discuss potential applications of this dataset in model validation, network analysis, and verification of algorithmic approaches such as community detection, link prediction, and network dismantling.

Strengths

i. Novelty and Scale: The dataset’s breadth, covering multiple years and large spatial scales, fills a significant gap in air traffic research resources, supporting macro-scale analyses.

ii. Comprehensiveness: The detailed multilevel network representations include temporal, spatial, and topological information, enabling diverse applications.

iii. Open Data and Reproducibility: The authors make the dataset openly available, facilitating transparency, reproducibility, and further research utilization.

iv. Potential for Diverse Applications: The discussion outlines numerous research avenues, including validation of complex network algorithms, workload assessment, and air traffic system resilience.

Weaknesses and Major Concerns

i. Methodological Clarity and Data Processing Details: The manuscript would greatly benefit from a clearer, more detailed description of the data processing pipeline specifically, how trajectories are converted into interaction networks, threshold criteria for defining interactions, and any preprocessing steps. Inclusion of flowcharts, pseudocode, or detailed algorithms would enhance reproducibility.

ii. Validation and Quality Assessment: While extensive, the paper lacks a rigorous validation of the interaction data for instance, an analysis of false positives/negatives in interaction detection or comparison with ground-truth data (if available). Clarifying the accuracy of these interaction specifications is necessary.

iii. Sensitivity Analysis: There is limited discussion on how choices such as interaction thresholds, spatial/temporal resolutions, or filtering criteria impact the network properties. Sensitivity analyses would strengthen confidence in the robustness of the dataset.

iv. Computational Cost and Scalability: Addressing the computational efficiency of data processing, as well as potential limitations in scaling or applying the approach to larger datasets, is crucial for understanding practical usability.

v. Application Demonstration: While the paper discusses potential applications, incorporating a preliminary case study or example analysis (e.g., comparison of network topology during different traffic conditions) would exemplify the dataset’s utility.

Minor Comments

i. Clarify whether the trajectory data filtering employs any smoothing or noise reduction techniques.

ii. Provide more details on the metadata associated with each interaction network, especially regarding the airspace status layers.

iii. Consistently refer to the dataset’s spatial and temporal resolution parameters used throughout.

iv. Enhance figures with visual representations of the network structure evolution over selected periods.

Recommendations

Major Revision: There is need to provide substantial improvements in methodological transparency particularly detailed descriptions of data processing algorithms, validation measures, and sensitivity analyses. Including a case demonstration to showcase the dataset’s analytical potential would further amplify the manuscript’s impact. Addressing these aspects will enhance reproducibility, clarity, and community adoption of this valuable resource.

Reviewer #3: This paper presents an unprecedentedly large dataset of aircraft interaction networks in European airspace. The dataset consists of 1708 daily networks of flight interactions between 2015 and 2021 and has been processed and validated with high accuracy.

These data can be used for air traffic dynamics analysis, testing of complex network algorithms, and large-scale epidemiological studies. The paper is excellent in terms of scale, accuracy, and accessibility, and is highly suitable for publication.

It is suggested that the practical applications of this dataset in air traffic management be briefly mentioned in the introduction section.

It is better to include a quantitative comparison with similar datasets (e.g. in terms of size, complexity, or spatio-temporal coverage) in the results section.

Adding metrics such as spatial accuracy, error rate, or compliance with operational data can increase the validity of the dataset.

Better insight into the network structure can be provided by analyzing features such as Modularity, Centrality, or Resilience.

It is better to clearly state limitations such as the exclusion of short flights or the impact of interpolation on accuracy.

Adding a small case study shows how this data can be used in practice.

6. PLOS authors have the option to publish the peer review history of their article (what does this mean?). If published, this will include your full peer review and any attached files.

Reviewer #1: No

Reviewer #2: No

Reviewer #3: No

---

## [Author Response · Author response to Decision Letter 1]

24 Oct 2025

---Reviewer 1---

>The paper introduces a unique, large-scale dataset of aircraft interaction networks spanning multiple years (2015–2021) and across the entire European airspace, which is unprecedented in scope. Though there are few points authors should address:

>1) While the dataset's utility is clear, the introduction could better highlight specific gaps in existing resources that this dataset uniquely fills.

We agree with the referee on the fact that the comparison was not clear. We have included in the introduction the only other data set publicly available presenting similar information, and described in (Gaume et al., TRIP, 2025). We also explicitly state the differences: they include information for approximately 11 months of 2022, and one Air Control Center (a region of France around Bordeaux), while we cover multiple years of the whole European airspace.

>2) Why the data is restricted to only four months?

>3) Whether the restriction to only four months per year won’t bias results.

>The implications of this sampling choice could be discussed more explicitly.

First of all, we would like to clarify that the decision of using four months is not ours: it is a limitation of the data provider, i.e. of EUROCONTROL. Notably, they do not state this in the main dataset website (https://www.eurocontrol.int/dashboard/aviation-data-research), but is then described in the accompanying documentation (see https://www.eurocontrol.int/sites/default/files/2025-04/eurocontrol-aviation-data-repository-research-metadata.pdf). We have made attempts at getting a more complete data set, but without success so far; this seems to be a legal limitation at their end.

Note that this is already mentioned in the section “Raw trajectory data pre-processing”: Data availability is constrained at the source to four months - March, June, September, and December.

As far as the implications and biases of this sampling, we suspect that they are minimal. The provided months cover very different conditions (from low traffic in December, to high traffic in June and September). They also provide examples of holiday periods, as e.g. of Christmas, when traffic is strongly reduced. Still, it is clear that the absence of a bias can only be proven by having access to all months, something impossible at present. In the event that EUROCONTROL changes this policy and publishes more information, we will of course update the networks accordingly.

>4) Proper qualitative, quantitative validation of the data should be provided including some visual presentation.

While we agree with the referee on the need of validation, the problem we face is that no ground-truth data are available for this. As we explain in the paper, due to confidentiality and security issues, full data about aircraft interactions are not disclosed - only some high-level statistics are available. This was precisely the motivation behind this analysis: we wanted to have a coherent data set of interactions, to support further studies. At the same time, this state of affairs also precludes a direct validation.

Due to this, we have opted for a qualitative validation, by: verifying the correctness of each step of the reconstruction; comparing our estimation of interactions with available statistics; and performing a statistical study of the results, to confirm that no bias is affecting the reconstruction. In the revised version of the text, this last part has especially been extended, by including statistics for individual days and the evolution throughout time - see for instance Figs. 4, 5 and 6.

>5) The paper could include a brief demonstration of a real-world analysis workflow such as building a temporal network analysis from raw CSVs.

We would like to highlight that this was already included in the initial version of the paper. Specifically, the Zenodo repository also stores several files illustrating how to load the data, and how to reproduce some of the topological analyses presented in the paper; we have highlighted this at the end of the “Usage notes” section. On the other hand, we have opted against including the code directly in the manuscript, in order to keep the focus of the main text on the methodology and the potential applications.

---Reviewer 2---

>i. Methodological Clarity and Data Processing Details: The manuscript would greatly benefit from a clearer, more detailed description of the data processing pipeline specifically, how trajectories are converted into interaction networks, threshold criteria for defining interactions, and any preprocessing steps. Inclusion of flowcharts, pseudocode, or detailed algorithms would enhance reproducibility.

We have taken steps towards improving the clarity of the description of the data processing pipeline. Specifically, we have added a new figure with the flowchart of the pipeline, illustrating the sequence of steps, and their organisation into trajectory- and interaction-based ones (Fig. 1). We have further added an additional figure with a synthetic example of how interaction networks are reconstructed (Fig. 2). Finally, some texts have been rephrased, to make them easier to follow.

>ii. Validation and Quality Assessment: While extensive, the paper lacks a rigorous validation of the interaction data for instance, an analysis of false positives/negatives in interaction detection or comparison with ground-truth data (if available). Clarifying the accuracy of these interaction specifications is necessary.

We completely agree with the referee on this point. Yet, the problem is both simple to state and very difficult to solve: no ground data are available. As we explain in the paper, due to confidentiality and security issues, full data about aircraft interactions are not disclosed - only some high-level statistics are available. This was precisely the motivation behind this analysis: we wanted to have a coherent data set of interactions, to support further studies. At the same time, this state of affairs also precludes a direct validation.

As detailed in the paper, we have opted for an indirect validation, by: verifying the correctness of each step of the reconstruction; comparing our estimation of interactions with available statistics; and performing a statistical study of the results, to confirm that no bias is affecting the reconstruction. In the revised version of the text, this last part has especially been extended, by including statistics for individual days and the evolution throughout time - see for instance Figs. 4, 5 and 6.

>iii. Sensitivity Analysis: There is limited discussion on how choices such as interaction thresholds, spatial/temporal resolutions, or filtering criteria impact the network properties. Sensitivity analyses would strengthen confidence in the robustness of the dataset.

As highlighted by the reviewer, there are two main factors affecting the identification of interactions.

On the one hand, one finds the thresholds (both horizontally and vertically) used to define when an interaction takes place. Our approach has been to include as many interactions as possible, for then leaving to the user the possibility of filtering them. To illustrate, each interaction also includes the minimal horizontal separation that both aircraft have achieved; hence, networks can easily be filtered, for instance to include only interactions below five nautical miles.

On the other hand, the filtering applied on the raw trajectories can also have an impact. We have analysed how many interactions are lost by excluding the trajectories that have been filtered out, and found a maximum of 14 per day (average of 0.9) - full statistics have now been included in the section “Raw trajectory data pre-processing”. This is to be expected, as by definition, those trajectories were extremely short and/or on the border to the considered airspace. In short, the impact of this filter is negligible. Still, we opted in favour of deleting those trajectories, as limited data are available for them, and may therefore be unreliable.

Finally, the reviewer is mentioning the temporal resolution of the data. This can also be understood from two perspectives. Firstly, the temporal resolution of the EUROCONTROL's data set, over which we have no control - some statistics are reported below. Secondly, the temporal resolution of the trajectory interpolation; in this case, we have opted for a very fine-grain representation (every second), in order to make sure that no interactions are missed.

>iv. Computational Cost and Scalability: Addressing the computational efficiency of data processing, as well as potential limitations in scaling or applying the approach to larger datasets, is crucial for understanding practical usability.

As suggested, we have added a new sub-section with an analysis of the computational cost. We have focused on two cases. On one hand, we report the cost associated to the reconstruction of the networks themselves; note that this is mostly relevant for us, as it quantifies the complexity of expanding the data set, but not its use. On the other hand, we also provide estimations of the cost of calculating two classical topological metrics, i.e. the modularity and efficiency.

>v. Application Demonstration: While the paper discusses potential applications, incorporating a preliminary case study or example analysis (e.g., comparison of network topology during different traffic conditions) would exemplify the dataset’s utility.

As suggested by the referee, we have added a new section with an example of an air traffic control application, namely “Analysis of daily interaction networks”. In it, we show how different interaction patterns emerge as a function of the total flown distance, and most importantly, of the day of the week. We hypothesise that the appearance of a higher demand for touristic destinations during the weekend may have a major impact in air traffic control. Note that these results are new, and expand over what already presented in two previous papers - which have been referenced.

>Minor Comments

>i. Clarify whether the trajectory data filtering employs any smoothing or noise reduction techniques.v. Application Demonstration: While the paper discusses potential applications, incorporating a preliminary case study or example analysis (e.g., comparison of network topology during different traffic conditions) would exemplify the dataset’s utility.

We have clarified, in the section “Raw trajectory data pre-processing”, that no smoothing nor noise reduction algorithms have been used. The rationale is that we assume trajectories to have a good quality; these algorithms would indeed be needed if the starting point were ADS-B data.

>ii. Provide more details on the metadata associated with each interaction network, especially regarding the airspace status layers.

We respectfully do not understand what the reviewer is referring to. Information about the metadata provided for each airspace region, i.e. the fields listed in Tabs. 4 and 5, is already included in the text, specifically in the paragraph starting with “We complement the above information...”, and with more than half page of length (in section “Data records”). Also note that some of the reported metrics, e.g. the entropy of the traffic, are only described here, but their exact definition is included in other papers - which are now more clearly referenced. Still, while checking the text, we have detected a couple of missing details in Tab. 4, which we have now completed.

In the event we have misunderstood the comment of the referee, we will be glad to consider any further suggestion.

>iii. Consistently refer to the dataset's spatial and temporal resolution parameters used throughout.

As suggested, we have added an analysis of the EUROCONTROL's data set temporal resolution, understood as the average time between consecutive position reports - see the section “Raw trajectory data pre-processing”. As for the spatial resolution, aircraft positions are reported as pairs of latitude and longitude, both with a resolution of one second of a degree - this is equivalent to ≈ 15 meters, depending on the latitude. While this is the theoretical resolution, we have no information about the kind of course-graining or other preprocessing steps that may have been performed on the trajectories. Note that this same spatial resolution is used to report interactions.

>iv. Enhance figures with visual representations of the network structure evolution over selected periods.

We have enhanced several representations. To illustrate, we have now added an illustration of how interactions and their associated networks are reconstructed (Fig. 2). We have further improved the analysis of the topology of networks, both in a single day, and the evolution throughout time (see Figs. 5, 6 and 7).

---Reviewer 3---

>It is suggested that the practical applications of this dataset in air traffic management be briefly mentioned in the introduction section.

We would like to highlight that these are already included in the introduction; in fact, the first half of the last paragraph of the introduction is devoted to them - see sentence “The contributed networks find a natural application in many problems within the context of air traffic management and control.” and following ones. Among others, these include the study of the propagation of interactions (for which models have been proposed, but that usually rely on synthetic data); the analysis of ATCO workloads; and the assessment of conflict resolution algorithms. In addition, we have added a section with some novel results, showing how the topology of the network is modified, possibly by shifts in demand, during weekends.

>It is better to include a quantitative comparison with similar datasets (e.g. in terms of size, complexity, or spatio-temporal coverage) in the results section.

It has to be noted that the only other data set publicly available presenting similar information is the one referenced in (Gaume et al., TRIP, 2025). This was not clear in the previous version of the paper, and has now been clarified in the introduction. We now also explicitly state the differences: they include information for approximately 11 months of 2022, and one Air Control Center (i.e. a region of France around Bordeaux), while we cover multiple years of the whole European airspace.

As an additional note: at submission, the available version of (Gaume et al., TRIP, 2025) was a conference proceedings with information for only approximately one month of operations. We have updated the text to reflect what the authors have documented in the updated version of the paper.

>Adding metrics such as spatial accuracy, error rate, or compliance with operational data can increase the validity of the dataset.

The referee is here highlighting an interesting point, which has also been raised by the other reviewers: how to validate the quality of the interactions we extracted. The main problem here is that no ground-truth data are available to perform such validation. As we explain in the paper, due to confidentiality and security issues, full data about aircraft interactions are not disclosed - only some high-level statistics are available. This was precisely the motivation behind this analysis: we wanted to have a coherent data set of interactions, to support further studies. At the same time, this state of affairs also precludes a direct validation. Note that in the manuscript we describe a more indirect validation, which also includes comparing our estimation of interactions with available statistics about losses of separation, as provided by EUROCONTROL, presented in the section “Technical validation”.

>Better insight into the network structure can be provided by analyzing features such as Modularity, Centrality, or Resilience.

As suggested, we have added an extensive analysis of the topology of the networks, both in a single day, and the evolution throughout time (Figs. 5 and 6). In addition, we have included a use case with an analysis of how these structures are dependent on operational factors, thus connecting to the air transport field (see section “Analysis of daily interaction networks”).

>Adding a small case

---

## [Decision Letter · Decision Letter 1]

2 Nov 2025

A large-scale data set of aircraft interaction networks.

PONE-D-25-41897R1

Dear Dr. López-Martín,

We’re pleased to inform you that your manuscript has been judged scientifically suitable for publication and will be formally accepted for publication once it meets all outstanding technical requirements.

Kind regards,

Absalom El-Shamir Ezugwu, Ph.D

Academic Editor

PLOS ONE

Additional Editor Comments (optional):

All concerns raised by the reviewers, who are experts in the field, have been carefully and satisfactorily addressed by the authors. I therefore recommend that the paper be accepted in its current form. However, the authors are advised to carefully proofread the final version to ensure it is free from grammatical and typographical errors.

Reviewers' comments:

Reviewer's Responses to Questions

**Comments to the Author**

1. If the authors have adequately addressed your comments raised in a previous round of review and you feel that this manuscript is now acceptable for publication, you may indicate that here to bypass the “Comments to the Author” section, enter your conflict of interest statement in the “Confidential to Editor” section, and submit your "Accept" recommendation.

Reviewer #2: All comments have been addressed

Reviewer #3: (No Response)

2. Is the manuscript technically sound, and do the data support the conclusions?

Reviewer #2: Yes

Reviewer #3: (No Response)

3. Has the statistical analysis been performed appropriately and rigorously?

Reviewer #2: Yes

Reviewer #3: (No Response)

4. Have the authors made all data underlying the findings in their manuscript fully available?

Reviewer #2: Yes

Reviewer #3: (No Response)

5. Is the manuscript presented in an intelligible fashion and written in standard English?

Reviewer #2: Yes

Reviewer #3: (No Response)

6. Review Comments to the Author

Reviewer #2: Dear Authors,

Thank you for your valuable response and contributions. All comments have been addressed

Reviewer #3: (No Response)

7. PLOS authors have the option to publish the peer review history of their article (what does this mean?). If published, this will include your full peer review and any attached files.

Reviewer #2: No

Reviewer #3: No

---

## [Editor Report · Acceptance letter]

PONE-D-25-41897R1

PLOS ONE

Dear Dr. López-Martín,

I'm pleased to inform you that your manuscript has been deemed suitable for publication in PLOS ONE. Congratulations! Your manuscript is now being handed over to our production team.

Kind regards,

on behalf of

Professor Absalom El-Shamir Ezugwu

Academic Editor

PLOS ONE